# Increased Dissemination of Aflatoxin- and Zearalenone-Producing *Aspergillus* spp. and *Fusarium* spp. during Wet Season via Houseflies on Dairy Farms in Aguascalientes, Mexico

**DOI:** 10.3390/toxins16070302

**Published:** 2024-07-01

**Authors:** Erika Janet Rangel-Muñoz, Arturo Gerardo Valdivia-Flores, Carlos Cruz-Vázquez, María Carolina de-Luna-López, Emmanuel Hernández-Valdivia, Irene Vitela-Mendoza, Leticia Medina-Esparza, Teódulo Quezada-Tristán

**Affiliations:** 1Departamento de Ciencias Veterinarias, Centro de Ciencias Agropecuarias, Universidad Autónoma de Aguascalientes, Aguascalientes 20131, Mexico; janet.rangel@edu.uaa.mx (E.J.R.-M.); carolina.deluna@edu.uaa.mx (M.C.d.-L.-L.); emmanuel.hernandez@edu.uaa.mx (E.H.-V.); teodulo.quezada@edu.uaa.mx (T.Q.-T.); 2División de Estudios de Posgrado e Investigación, Tecnológico Nacional de México, Instituto Tecnológico El Llano Aguascalientes, km 18 carretera Aguascalientes—San Luis Potosí, El Llano, Aguascalientes 20330, Mexico; cruva18@gmail.com (C.C.-V.); irene.vm@llano.tecnm.mx (I.V.-M.);

**Keywords:** mycotoxins, prevention, farm animals, climate, insects, mycotoxin control

## Abstract

Crops contamination with aflatoxins (AFs) and zearalenone (ZEA) threaten human and animal health; these mycotoxins are produced by several species of *Aspergillus* and *Fusarium*. The objective was to evaluate under field conditions the influence of the wet season on the dissemination of AF- and ZEA-producing fungi via houseflies collected from dairy farms. Ten dairy farms distributed in the semi-arid Central Mexican Plateau were selected. Flies were collected in wet and dry seasons at seven points on each farm using entomological traps. Fungi were isolated from fly carcasses via direct seeding with serial dilutions and wet chamber methods. The production of AFs and ZEA from pure isolates was quantified using indirect competitive ELISA. A total of 693 *Aspergillus* spp. and 1274 *Fusarium* spp. isolates were obtained, of which 58.6% produced AFs and 50.0% produced ZEA (491 ± 122; 2521 ± 1295 µg/kg). Houseflies and both fungal genera were invariably present, but compared to the dry season, there was a higher abundance of flies as well as AF- and ZEA-producing fungi in the wet season (*p* < 0.001; 45.3/231 flies/trap; 8.6/29.6% contaminated flies). These results suggest that rainy-weather conditions on dairy farms increase the spread of AF- and ZEA-producing *Aspergillus* spp. and *Fusarium* spp. through houseflies and the incorporation of their mycotoxins into the food chain.

## 1. Introduction

Filamentous fungi can produce several low-molecular-weight compounds, called secondary metabolites, which exert biological functions in fungal competition and survival within ecological niches; some of these secondary metabolites are considered mycotoxins (MTs) because they can induce toxic effects in both humans and animals. Some of the major MTs of the genera *Aspergillus* and *Fusarium* are aflatoxins and zearalenone (AFs and ZEA), respectively [1]. In general, the incorporation of toxigenic fungi and their MTs into the food chain is initiated via the dissemination of their spores, infection and propagation in plant tissues, and mycelial growth during the development of agricultural crops and in the storage or final processing of harvested products [2,3]. The MTs are not usually destroyed by boiling, nor do they confer taste, color, aroma, or appearance to contaminated ingredients, so they usually go unnoticed by both humans and animals [2,4]. In addition, the presence of mycotoxins in the food chain produces a strong economic impact associated with the contamination of agricultural crops and food of animal origin, deterioration of animal health, and decrease in their reproduction and productive behavior [5,6].

The AFs are synthesized with nearly 20 *Aspergillus* species, mainly *A. flavus*, *A. nomius*, and *A. parasiticus* [3,4]. Weather conditions modify *Aspergillus* spp. growth dynamics, sporulation, and AF biosynthesis pathways; *Aspergillus* spp. have a wide range of conditions necessary for their growth; however, their ability to produce AFs is more intensely expressed in humid and warm conditions (0.96–0.99 a_w_, 25–30 °C) [7,8,9,10,11]. *Aspergillus* spp. are distributed worldwide, especially in tropical and subtropical climates where they contaminate grains, seeds, oilseeds, and livestock fodder [2,12]. The nutritional basis on dairy farms of the herd throughout the agricultural year is the fiber–energy–protein feed rations made from grains, oilseed by-products, and corn silage harvested and processed precisely during the wet season; however, these ingredients are usually contaminated with both *Aspergillus* spp. and AFs [13,14]. AFs are also in bovine dairy feed and feed components, with their estimated prevalence being 54–64% across the globe; although the average global concentrations are relatively low (2.6–3.9 μg/kg), in tropical and subtropical countries, these can usually be five or ten folds higher [13,14,15,16].

AFs are bifuran–coumarin compounds. There are four primary forms of AFs naturally present in agricultural products (AFB_1_, AFB_2_, AFG_1_, and AFG_2_); the other forms of AFs are derived from the metabolic processing of these primary forms within the human or animal body [4]. AFs are among the most potent mycotoxins and their exposure has been linked to human hepatocellular carcinoma, growth impairment, immune system dysfunction, and acute toxicity; AFs also cause a range of adverse effects in animals [2,17]. In dairy cows, exposure to usually low doses of AFs affects liver function and immune capacity, causing susceptibility to disease, reducing milk production, and increasing culling rate [16,18].

When dairy cows ingest forage contaminated with AFs, these MTs are absorbed into the digestive tract, distributed throughout the bloodstream, and processed mainly in the liver and kidneys, where they generate intermediate chemical compounds as hydroxylated metabolites (AFM_1_ and AFM_2_) that can be transferred to milk and also exert their toxic effects and carcinogenic or immunosuppressive damage to the humans and animals that consume them [17]. The AFM_1_ contamination of raw milk has been observed in some regions such as the Mexican Central Plateau; this contamination shows a seasonal pattern during the wet period and the proportion of raw milk samples exceeding the applicable maximum residues limit (MPL: 0.05 µg/kg) is more than three times higher than during the dry period [14,19,20,21,22].

On the other hand, ZEA is a toxic secondary metabolite synthesized via several *Fusarium* species, especially *F. graminearum*, *F. cerealis* (*F. crookwellense*), *F. equiseti*, *F. culmorum*, *F. verticillioides*, and *F. semitectum* [23,24]. The ability to produce ZEA varies among species and strains of the same species; moreover, the amount of ZEA formed depends on environmental factors [24]. The highest amount of ZEA produced has been observed at conditions above 25 °C and 16% humidity, in particular with conditions of neutral or acidic pH and high CO_2_ concentrations [25,26]. The occurrence of ZEA has been identified worldwide in many cereals; maize and wheat have a high prevalence of ZEA contamination; however, rice, barley, and oats have also been found to be occasionally contaminated with this toxin [27,28].

When ZEA is ingested in contaminated diets, ZEA is rapidly absorbed through the gastrointestinal tract and dispersed widely in body tissues. The liver is the main organ of the distribution of ZEA, followed by a slow elimination rate and extensive enterohepatic recirculation [25]. In dairy cows, the derivative zearalenol (ZEL, α, β) is generated in the liver and can be transferred to milk and dairy products to be ultimately ingested by humans [5,29]; three other metabolites of ZEA are termed α-zearalanol, β-zearalalanol, and zearalanone. ZEA exhibits low acute toxicity (oral LD_50_ 4–20 g/kg) in laboratory animals; however, long-term exposure can cause significant health risks [26,29,30].

The chemical structure of ZEA (resorcyclic acid lactone) has similarities to natural estrogens, such that it can competitively bind to the cellular receptor and exhibit non-steroidal estrogenic properties. ZEA exerts immunotoxicity, hepatotoxicity, nephropathy, hematotoxicity, and alterations in endocrine and lipid metabolism; ZEA especially alters lipid-related signaling, thereby inducing significant changes in circulating adipokine concentrations [31,32]. ZEA has been associated with reprotoxic xenoestrogens, endometrial dysfunction, and mammary carcinomas in women who ingest it in contaminated food [25,33,34]. In non-ruminant animals, ZEA causes hepatotoxicity; nephrotoxicity; hematotoxicity; immunotoxicity; impaired blood coagulation; and decreased sperm count, serum testosterone level, and fertility [30,35]. In dairy cattle, ZEA causes changes in estrogenic activity, reproductive problems, and infertility, consequently inducing a decrease in milk production and significant economic losses [29,35,36,37].

Mycotoxins have reached a special interest in dairy cattle due to their negative effect on public health triggered by the presence of their metabolites in milk and dairy products [38]. To reduce the problems caused by mycotoxins, maximum permissible limits (MPLs) have been established. In Mexico, MPLs have been established for AFs in cereals and AFM_1_ in milk (20 and 0.5 µg/kg); however, it is common for the dairy industry to adhere to international standards for AFM_1_ (0.05 µg/kg), ZEA, and other mycotoxins [19,20,21,22]. For ZEA, there is a great diversity in legislation; the European Commission has limited the ZEA content in dairy feed or feed materials to 500 or 2000 µg/kg, except that in maize, the proposed maximum limit is 3000 µg/kg [38,39]. In fact, when feed contamination with AFs or ZEA occurs, it is because the conidia of *Aspergillus* spp. or *Fusarium* were able to reach a substrate in suitable conditions for their development; that means that contamination with spores is a necessary condition for the contamination of the food chain, which needs to be complemented with the lack of good agricultural or livestock practices. Therefore, the dissemination of conidia via vectors such as houseflies can be a critical factor for contamination control.

The housefly (*Musca domestica* L.) is a cosmopolitan insect with high gregarious, feeding, and sexual activity; it constitutes a pest closely related to human and agricultural activities and a threat to public and animal health [40,41]. In their larval stage, flies are nutritionally dependent on microbes existing in decaying organic substrates, with relationships that may transcend their larval stages of development [42,43]. Nutritional dependence diversifies extensively when houseflies reach the adult stage, whereby they energetically seek to reach manure, animal exudates, food, and other organic substrates for feeding or reproducing [40,44]. Decaying organic substrates are rich in bacteria and other microbes, which electrostatically adhere to the adult fly’s exoskeleton, especially its adhesive parts [28,45], and to the inside of its digestive tract so that when its body surfaces, saliva or feces comes into physical contact with other materials and a pathway for the dissemination of prevalent and emerging biological agents is established [46,47,48].

The microbial load that adult flies are capable of disseminating is very large. More than 200 pathogens have been identified that are spread by flies; flies can spread pathogens over distances as far as 100 km; plus, each fly can harbor up to a hundred pathogenic microbes on or inside its body [40,44]. The microbiota detected in flies is very diverse but is predominantly composed of bacteria [44,49]; although less studied, the housefly has also been detected to be a pathway for the dissemination of filamentous fungal spores [50,51,52]. On the other hand, meteorological conditions have also been shown to modify the abundance and mobility of the housefly [46]; although the biological cycle is completed in three weeks, in humid and warm seasons (60–70% relative humidity; 25–35 °C) and with nutrient-rich substrates, it can be reduced to as little as 8 days, which allows it to exponentially accelerate population growth [40,42,44].

Although the impact of climate on housefly activity and separately on fungal development is well known, simultaneous effects on both organisms—additive or synergistic—are scarce or inconsistent [53,54,55]. From the above, it is assumed that the seasonal increases in precipitation and humidity, under stable conditions of suitable temperatures, exert synergistic relationships between the increased development of toxigenic fungi and housefly activity, expanding the dissemination of fungal spores, infection, and spread to plant tissues; mycelial growth in agricultural crops; and worsening the contamination of feed and dairy products with mycotoxins.

The objective of this study was to evaluate under field conditions the influence of different precipitations, humidity patterns, and temperatures in the wet season on the dissemination of AF- and ZEA-producing *Aspergillus* spp. and *Fusarium* spp. via houseflies collected from dairy farms.

## 2. Results

### 2.1. Fungal Mycobiota and Housefly Population

A total of 5051 fungal isolates were identified from flies collected in both the dry and wet seasons; of these isolates, morphological characteristics congruent with the genera *Fusarium*, *Mucor*, *Aspergillus*, *Rizhopus*, *Penicillium*, *Cladosporium*, *Absidia*, *Cunninghamella*, *Eurotium*, *Geotrichum*, and *Alternaria* were identified (25.3, 24.6, 13.7, 13.5, 10.6, 2.7, 2.5, 1.9, 1.9, 1.6, 1.5%, respectively). Table 1 shows a comparison of the average number of houseflies that were captured in the dry and wet seasons, within each of the seven sites of the ten participating dairy farms, as well as the average number of fungal isolates obtained from the flies at each sampling point. Compared to the dry season, during the wet season, there was a highly significant (*p* < 0.001) increase in the average number of flies collected (231/45.3 = 5.1 times greater); simultaneously, the average number of fungal isolations per trap increased (68.3 ± 14.0 vs. 3.9 ± 0.6). Thus, in the wet season, the average number of fungal mycobiota increased 17.5-fold and this increase meant that, in addition to the increase in the number of flies, the fungal load per fly also increased since during the dry period, 8.6 out of every 100 flies were contaminated with some fungus (3.9/45.3), while the proportion of contaminated flies tripled to 29.6% during the wet period (68.3/231).

The increase in fungal mycobiota and houseflies was not uniform across all participating dairy farms. The efficiency in the application of fly control methods (pyrethroid insecticide application, feed ingredient protection, manure collection, solarization, final disposal, etc.) was associated with fly abundance, which in turn had an impact on the increase in fungal dissemination in the body of the flies. On the other hand, flies were distributed homogeneously among the different sites of each farm; although there was a predilection for sites with higher affluence to have more nutritious feed substrates (silo-cutting surface, feed store, etc.), the variation associated with the efficiency of the different fly control methods increased the variation detected in the number of collected flies and prevented reaching statistical significance (*p* > 0.05) among the means of the sampled sites.

### 2.2. Toxigenic Aspergillus spp. and Fusarium spp.

Table 2 shows a comparison between the average number of *Aspergillus* spp. and *Fusarium* spp. obtained from the flies captured at each sampling point in the dry and wet seasons; the ability to produce AFs or ZEA under in vitro conditions is also analyzed. A total of 693 fungal isolates showed morphological characteristics congruent with the genus *Aspergillus* and 1274 with the genus *Fusarium.* A total of 406 isolates of *Aspergillus* spp. with AF production capacity were obtained—385 in the wet period and 21 in the dry period. Also, 637 ZEA-producing *Fusarium* spp. were obtained (504 during the wet period and 133 in dry the period). A high potential for *Aspergillus* and *Fusarium* fungi isolated from houseflies to produce AFs and ZEA was estimated. The production of AFs in *Aspergillus* fungi in both seasons was estimated at average concentrations of 491 ± 122 µg/kg of fungal mass and growth medium (range: 127–2376 µg/kg), while the ZEA production capacity of *Fusarium* isolates was estimated at 2521 ± 1295 µg/kg (range: 17.5–38,591 µg/kg). In general (Figure 1), a highly significant (*p* < 0.001) increase in the AF- and ZEA-producing fungal load on the fly population was observed in the wet season (5.5/0.3 = 18.3 and 7.2/1.9 = 3.8, respectively), as well as a considerable increment in the proportion of toxigenic isolates of the genera *Aspergillus* and *Fusarium*, which doubled compared to the dry season (79.3/32.9 = 2.4 and 42.5/22.6 = 1.9%, respectively).

Table 3 shows a general linear model analysis to characterize both the individual significance (*p*-value) of each of the factors that could be interacting to influence the mycobiota and the housefly population as well as the combined determination of the variation (coefficient of determination R^2^) in the number of flies and fungi. Overall, the increase in mycobiota and AF- and ZEA-producing *Aspergillus* spp. and *Fusarium* spp. during the wet season was significantly (*p* < 0.001) related to the combined effects of pluvial precipitation, relative humidity, and general characteristics inherent to the wet season (R^2^: 46.4–74.4%; Table 3). Also, the number of isolates obtained from *Aspergillus* and *Fusarium* genera, as well as the number of these isolates that could produce AFs and ZEA, was significantly (*p* < 0.001) associated with the number of trapped houseflies (correlation coefficient 0.630–0.887). In the general linear model analysis, dairy farm sites, nested within the same dairy farm, were only significant for the number of ZEA-producing *Fusarium* spp. isolates. Other factors, such as mean, maximum, and minimum temperatures, had no significant direct influence (*p* > 0.05) but were independently related to seasonal meteorological variables, such as rainfall and relative humidity.

Consistent with the increase noted for flies and fungi, AF and ZEA contamination in the total mixed ration and AFM1 in raw milk were significantly higher in the wet season compared to the dry season. The total mixed ration used on dairy farms also showed significant differences (*p* < 0.05) in AF and ZEA contamination between the dry and wet seasons (18.4 ± 1.17 vs. 24.7 ± 2.7 and 147 ± 38.6 vs. 228 ± 41.4 µg/kg, respectively). AFM_1_ was also detected in raw milk in the dry and wet seasons (7.3 ± 2.4 vs. 24.1 ± 6.0 ng/kg, respectively). As compared to the dry season, the proportion of samples that exceeded the maximum residue limits allowed by the applicable legislation increased (*Px*^2^ < 0.01) during the wet season (AFs > 20 µg/kg: 66.7/30.8% = 2.2; ZEA > 100 µg/kg: 66.7/46.2% = 1.45 AFM_1_ > 50 ng/kg: 6.8/1.44% = 4.7).

## 3. Discussion

In this study, it was observed that compared to the dry season, the wet season’s environmental conditions were significantly (*p* < 0.001) associated with an increase in both the housefly and AF- and ZEA-producing fungal populations. The presence of toxigenic *Aspergillus* spp. and *Fusarium* spp. on dairy farms constitutes a relevant problem because the aflatoxins and zearalenone they produce are involved in the food chain and affect public health; in addition, these mycotoxins deteriorate animal health and production [5,6,25]. However, to our knowledge, information on the influence of meteorological conditions on the intensity of the process of dissemination of toxigenic fungi by houseflies is published here for the first time.

In previous studies developed in the Central Mexican Plateau, the fungi *Aspergillus* spp. and *Fusarium* spp. have been identified in various substrates and feeds or forage used on dairy farms [13,14,56]; meanwhile, other studies [50,51,52,57] have shown the process of fungal spore dissemination via the housefly. The present study adds evidence of the effect of rainy-weather conditions on the increase in fungal conidia dissemination. In addition, we identified the existence of a combined effect between fungal proliferation and simultaneous fly population growth. This effect was clearly observed with the increase of 5 times more flies and a toxigenic fungal load per fly 11 times higher in the wet season compared to the dry season. Therefore, the present study suggests a great impact of wet climatic conditions on the increased dissemination of fungal spores between different sites from dairy farms, independently of the type of organic matter found at that location. In addition, a large proportion of *Aspergillus* spp. and *Fusarium* spp. isolates showed the ability to produce high concentrations of AFs and ZEA in vitro. In the laboratory, over half of *Aspergillus* spp. isolates showed the ability to produce significant concentrations of AFs, while a third of *Fusarium* spp. isolates did so for ZEA, demonstrating their viability and toxigenic potential. It has already been shown that these fungi have a wide range of conditions necessary for their growth; however, their AF and ZEA production capacities are expressed more intensely under warm and humid conditions (25–28 °C, 0.96–0.99 aw) [8,9,11].

In this study, a highly significant (*p* < 0.001) increase in the total number of fungi on dairy farms was observed during the wet season compared to the dry season. It has been shown that when rainfall occurs with warm temperatures, suitable environments are established for fungal proliferation, mycotoxin production [6,55,58], and fly multiplication [59,60]. This also suggests that weather conditions during the wet period of this study influenced a higher abundance of toxigenic fungi and fly contamination compared to the dry period. In the present study, it was possible to isolate a significant number (5051) of fungi of various genera; this finding suggests that the broad presence of fungi and houseflies in the environment may play an important role in the dissemination of spores with an infective capacity. Several fungal genera have already been identified on the body surface of houseflies trapped in anthropogenic spaces [61,62,63]. These toxigenic fungi have also been detected in isolates obtained from flies caught in rural septic tanks [62,63].

The toxic, carcinogenic, immunosuppressive, and other negative effects of AFs and ZEA on human and animal health are well known [6,17,25]. These mycotoxins have been detected in concentrations of concern in food and feed in the Central Mexican Plateau, while other mycotoxins (FBs, DON, and OTA) showed concentrations below the MPLs [13,14,16,61]. The ability of toxigenic fungi to use fly bodies to spread their viable spores for mycelial development from different areas within dairy farms suggests that toxigenic fungi use the housefly as an imposed mechanical vector, which may imply the existence of a very active pathway of crop contamination in toxigenic fungi.

It has also been demonstrated that humidity significantly influences the proliferation of the housefly and other insects, even in climate change scenarios [45,48,62,63]. Therefore, it is important to consider that the wet season in the semi-arid climate of the study area favors the growth of fungi, the production of AFs and ZEA, and the synergistic effect with the increase in the population of mechanical vectors. In this study, it was also noted that both the number of isolates and the number of AF- and ZEA-producing *Aspergillus* spp. and *Fusarium* spp. were significantly correlated with the number of flies so that during the wet season, in addition to there being more flies, they were more contaminated by fungi. On the other hand, differences in housefly numbers were associated (*p* < 0.05) with the application of insect control measures. It has been noted that housefly abundance is associated with poor hygienic measures and the odorous attraction given off by decaying organic materials, being especially abundant when humid conditions are present [40]. Thus, as the affluence of flies increased, the potential for magnifying the on-farm dissemination of toxigenic fungi, as well as their negative effects on feed quality, was also increased.

In the present study, a large increase (3.5-fold) in the number of flies contaminated with fungi was observed during the wet period compared to the dry period. It was also noticeable that the average proportion of AF- and ZEA-producing *Aspergillus* spp. and *Fusarium* spp. contaminating the fly body doubled during the wet period. It has already been described that humid conditions in warm weather can also promote fungal growth and sporulation, increasing the availability of spores for flies to transport and disperse [53]. This suggests that high humidity levels may improve the quantity, availability, and viability of fungal spores by preventing their desiccation and solar exposure and increasing fly contamination and dissemination in the sites where they roam in search of food or ovipositing materials. The role of the housefly as a disperser of different pathogens, especially bacteria, viruses, and parasites, has already been widely demonstrated [46,47,59]. Thus, the isolation of AF- and ZEA-producing *Aspergillus* spp. and *Fusarium* spp. from the fly body strongly suggests that the housefly acts as a mechanical vector of the toxigenic fungi on dairy farms.

## 4. Conclusions

The present study suggests that during the wet season, toxigenic species of *Aspergillus* spp. and *Fusarium* spp. intensify the use of the fly as an imposed mechanical vector, which mobilizes the spores from the different natural reservoirs to the food substrates present in the sites where humans and domestic animals live; this may imply the existence of a fast pathway for fungal incorporation into crops as well as, ultimately, the aflatoxin and zearalenone contamination of the food chain, representing an important risk for human and animal health as well as for rural production and the economy of the dairy industry.

Therefore, it is necessary to design holistic strategies to control food contamination with mycotoxins in which, in addition to the usual pre-harvest and post-harvest control measures for feedstuffs, meteorological conditions and the presence of other routes of dissemination of toxigenic fungi must be considered; specifically, housefly control on dairy farms during the wet season is essential to reduce the spread of fungi to feed ingredients and to limit the contamination of feed and raw milk with their mycotoxins.

## 5. Materials and Methods

### 5.1. Study Design

The study was conducted in the dry and wet seasons (April 2022 and August 2022) with a descriptive, non-experimental design. Ten dairy farms were selected via the non-probabilistic method of convenience in the central valley of the state of Aguascalientes (21°48′ N, 102°03′ W; 1800–1920 masl) in the Central Mexican Plateau. This region presented a semi-dry temperate climate with summer rainfall [63,64].

All selected dairy farms had comparable zootechnical conditions, buildings, and facilities. Inspections were carried out at seven different sites on each farm (feeders of low, middle, and high milk production; rearing room; milking parlor; feed store; and silo-cutting surface); in brief, Holstein cows were housed in pens bounded by metal fences with free access to feeders. Cows were grouped according to the amount of milk produced (low: <18, middle: 18–25, and high: >25 kg/d) and received a diet formulated as a total mixed ration to meet the nutritional requirements according to their level of milk production. Calves were separated from the cow and were fed manually during the first two months in individual cages located in a rearing room. The milking process was carried out with automated equipment inside a milking parlor and the milk production was sent to regional agro-industrial factories. The farms had a feed store to stock the supplies used to prepare the rations; the rations were prepared by mechanically mixing an energy concentrate with corn silage harvested on the same farm during the wet season. The required amount of corn silage was obtained daily from one end of the silo, called the silo-cutting surface.

### 5.2. Meteorological Variables

The record of each climatic variable was obtained by accessing the database of the National Network of Automated Agrometeorological Stations through the National Laboratory of Modeling and Remote Sensing of the National Institute of Forestry, Agriculture and Livestock Research [55]. Data from agrometeorological stations close to the selected farms (<15 km) were selected. The following climatic variables were considered: pluvial precipitation, relative humidity, and average daily temperature (middle, maximum, and minimum). Each farm was monitored during the two seasons (Table 4) when, according to historical climatic data, weather conditions were most dissimilar [65].

### 5.3. Housefly Capture

Houseflies were trapped with single-use entomological traps, which were prepared according to the model employed by Phoku et al. (2017) [52]. Each trap included two compartments: one at the bottom (0.5 L) where a natural, commercial, non-toxic olfactory attractant (Emmett R, León, Guanajuato, Mexico) was placed, as well as another compartment at the top (1.0 L) with holes for ventilation, one-way funnel-shaped access, and a textile mesh at the base to separate it from the lower compartment. At each selected dairy farm site (Table 1), two traps were exposed for 24 consecutive hours. The traps with the captured flies were transported to the laboratory inside a refrigerated cooler and placed in a low-temperature environment (3–5 °C) [66]; they were counted, the number of flies captured per trap was recorded, and the average per farm, date, and capture site was calculated.

### 5.4. Fungal Isolation

Flies were externally disinfected with sodium hypochlorite (1%; 0.5:00 min) according to the method described by Sales et al. (2002) [60] and placed in sterile polypropylene tubes. Fungi were isolated from both the outside and inside of the flies’ bodies (Figure 1). The isolation of fungi contaminating the exterior of the body was performed using the direct seeding method with serial dilutions [51]; for this purpose, disinfected flies from each trap (0.30 ± 0.10 g) were transferred to sterile tubes containing peptonized water (0.1%) with Tween 20. The flies were washed via gentle agitation (5 min) and the suspension obtained was considered at a 1:10 dilution; successive decimal dilutions (10^−2^, 10^−3^, and 10^−4^) were then prepared for inoculation (0.100 mL) in duplicate in Petri dishes containing Sabouraud dextrose agar, to which chloramphenicol was added as a bacterial inhibitor (100 mg/L) [67,68]. Seeding was performed via diffusion using sterile glass beads; the inoculated Petri dishes were incubated (28 ± 1.0 °C) in the dark for 7 days.

The isolation of the fungi inside the body of the flies was performed in quadruplicate according to the wet chamber method [69]; for this, seven disinfected fly specimens were placed equidistantly in a polypropylene Petri dish (90 × 15 mm) with a double layer of filter paper moistened with distilled water. The humid chambers were monitored for 7 days until fungal growth was observed on the body of the flies (Figure 2). Subsequently, a portion of the mycelium was taken using a sterile dissection needle and transferred to a new Petri dish with potato agar and dextrose growth medium; the dishes were incubated in the dark (28 ± 1.0 °C) and monitored daily for seven days. Isolated colonies were prepared with cotton blue staining with lactophenol, and the slides were observed under an optical microscope (Axio Star Plus, Carl Zeiss, Werk Göttingen 37081, Göttingen, Germany) at different magnifications for taxonomic identification [70,71].

### 5.5. Quantification of Mycotoxins

The quantification of AFs and ZEA was performed in duplicate on monosporic cultures of *Aspergillus* spp. and *Fusarium* spp. (7 d) using a competitive enzyme-linked immunosorbent assay; the standardized protocols suggested by the manufacturer were followed (Ridascreen Fast: Aflatoxin, Zearalenone, and Aflatoxin M1; article number: R5202, R5502 and R1121; R-Biopharm, AG, Darmstadt, Germany). Complete monosporic cultures in their Petri dishes were dried in a forced air circulation oven and the amount of dry matter contained in the culture medium was estimated. Subsequently, the cultures were processed in a Dunce-type homogenizer and mycotoxin extraction was performed with a mixture of methanol and water (70% *v*/*v*). The absorbance quantification of each sample was read on a microplate reader (BioTek ELx800, BioTek Instruments, Winooski, VT, USA) at a wavelength of 450 nm. MTs were also quantified in raw milk and total mixed ration samples without binders, fungicides, or other additives. Three samples per dairy farm were obtained at each station and processed in duplicate as noted above using the enzyme-linked immunosorbent assay procedure (R-Biopharm, Germany, AG, Darmstadt, Germany).

Mycotoxin concentration was estimated with software (Ridasoft Windows version 1.8) to interpret the absorbance result and verified using calibration curves prepared with purified standards of each of the mycotoxins (AFB_1_, AFB_2_, AFG_1_, AFG_2_, AFM_1_, and ZEA; Sigma Aldrich, St. Louis, MO, USA). The limits of detection of AFs, ZEA, and AFM_1_ (LOD; 0.5, 2.5, and 0.005 µg/kg), detection ranges (LOD-40, LOD-40, and LOD-0.080 µg/kg), coefficients of determination (R^2^: 99.7, 92.8, and 91.4%) and coefficients of variation (7.7, 9.7, and 11.2%) were estimated with the calibration curves. When the concentration of MTs exceeded the maximum concentration of the detection range, the test was repeated with an appropriate dilution of the sample. All measurements were performed in the presence of purified MT standards. The concentration of ZEA and AFs was estimated from the combined weight of the mycelium and dry matter present in the culture medium.

### 5.6. Statistical Analysis

The average number of houseflies that entered the two traps placed at each dairy farm site over 24 h was considered as the unit of observation of this study. The average number of *Aspergillus* spp., *Fusarium* spp., and other fungi isolates obtained in each observation unit was also estimated according to the total number of replicates performed in the serial dilution and wet chamber methods. Data were analyzed using one-way ANOVA; the dairy farm, sampling location, and dry or wet season were considered as independent factors. The factors were tested for normality and homoscedasticity. The procedure for Tukey’s test of separation of means (honestly significant difference) was performed. A simple correlation analysis was performed between the number of AF- and ZEA-producing *Aspergillus* spp. and *Fusarium* spp. and the total number of fungal isolates per observation unit versus the total number of flies collected in each observation unit. A general linear model analysis with nested effects was also included to determine the influence of the predictor variables (season, rainfall, relative humidity, dairy farm, and capture site) on the abundance of fungi and flies; alternate models were compared and the value of the coefficient of determination (R^2^) was used to select the best model. Statistical software (Statgraphics Centurion XV version 16.1.03) was used in all analyses and a *p*-value < 0.05 was considered significant.

## Figures and Tables

**Figure 1 toxins-16-00302-f001:**
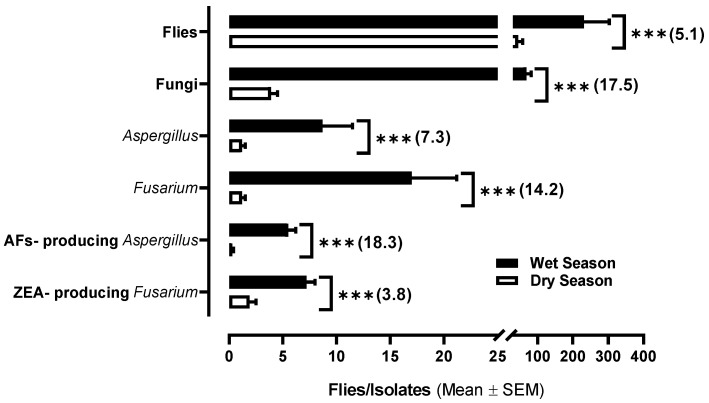
Capture of houseflies on dairy farms in the Mexican Central Plateau and fungal isolates in the dry and wet seasons. *** Statistically significant difference between seasons (*p*-value of <0.001). (Number): comparison of the means (wet/dry seasons).

**Figure 2 toxins-16-00302-f002:**
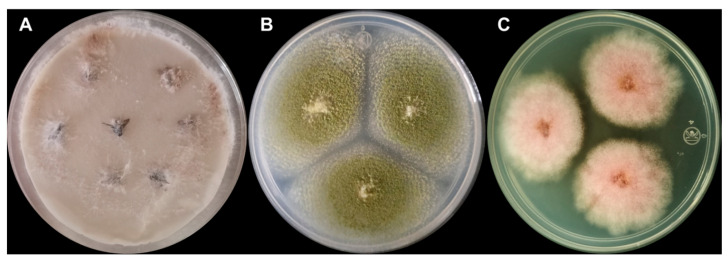
Isolation of fungi using the wet chamber method from externally disinfected houseflies. Panels: (**A**) mycelial growth on houseflies (*Fusarium* sp.; filter paper moistened with distilled water, 7 days); (**B**) isolation of *Aspergillus* sp. (PDA, 28 ± 1.0 °C, 7 days in the dark); and (**C**) isolation of *Fusarium* sp. (PDA, 28 ± 1.0 °C, 7 days in the dark).

**Table 1 toxins-16-00302-t001:** Houseflies captured in seven different sites of ten dairy farms located in the Mexican Central Plateau, as well as fungal isolates from the captured flies.

Location	Number ^1^	Dry Season ^2^		Wet Season ^2^	
		Flies	Fungi	Flies	Fungi
Dairy Farm					
I	7	6.3 ^c^ ± 0.6	2.0 ^b^ ± 0.49	17.1 ^d^ ± 1.4 **	4.1 ^c^ ± 0.3 ^ns^
II	7	6.4 ^c^ ± 0.7	0.6 ^b^ ± 0.19	20.7 ^d^ ± 2.0 **	4.3 ^c^ ± 0.7 *
III	7	23.4 ^bc^ ± 2.8	5.3 ^ab^ ± 0.59	72.6 ^cd^ ± 12.2 **	5.6 ^c^ ± 0.5 ^ns^
IV	7	19.3 ^bc^ ± 0.9	1.6 ^b ±^ 0.43	72.9 ^cd^ ± 8.5 ^ns^	10.0 ^c^ ± 1.2 **
V	7	15.3 ^c^ ± 1.7	3.6 ^b^ ± 0.54	88.4 ^cd^ ± 9.6 **	13.3 ^c^ ± 1.4 **
VI	7	25.6 ^bc^ ± 4.5	4.4 ^ab^ ± 0.64	125 ^cd^ ± 23.3 **	15.9 ^c^ ± 2.1 *
VII	7	49.4 ^c^ ± 6.7	2.4 ^ab^ ± 0.33	143 ^cd^ ± 12.4 **	42.3 ^c^ ± 3.8 **
VIII	7	178 ^a^ ± 10.3	5.1 ^ab^ ± 1.19	178 ^c^ ± 28.5 ^ns^	48.9 ^c ±^ 4.0 **
IX	7	19.0 ^bc^ ± 1.3	4.9 ^ab^ ± 0.61	551 ^b^ ± 40.7 **	156 ^b^ ± 8.6 **
X	7	111 ^b^ ± 16.8	8.7 ^a^ ± 1.19	1038 ^a^ ± 66.7 **	383 ^a^ ± 23.7 **
Site					
Feeders (Low)	10	38.2 ^b^ ± 11.6	2.5 ^b^ ± 0.5	172 ^a^ ± 61.2 *	56.2 ^a^ ± 12.0 *
Feeders (Middle)	10	50.9 ^b^ ± 13.5	2.8 ^b^ ± 0.6	238 ^a^ ± 66.0 **	48.8 ^a^ ± 9.1 *
Feeders (High)	10	43.6 ^b^ ± 9.9	3.4 ^b^ ± 0.5	233 ^a^ ± 60.6 **	69.2 ^a^ ± 11.8 **
Rearing room	10	47.1 ^b^ ± 14.0	1.6 ^b^ ± 0.4	285 ^a^ ± 74.8 **	63.0 ^a^ ± 9.7 **
Milking parlor	10	58.5 ^a^ ± 18.8	4.1 ^a^ ± 0.9	226 ^a^ ± 97.5 ^ns^	65.4 ^a^ ± 16.8 ^ns^
Silo-cutting surface	10	40.2 ^b^ ± 8.2	8.0 ^b^ ± 0.5	217 ^a^ ± 50.7 **	83.1 ^a^ ± 16.8 *
Feed store	10	38.6 ^b^ ± 14.2	4.6 ^a^ ± 0.6	242 ^a^ ± 92.0 *	92.4 ^a^ ± 21.7 *
All	70	45.3 ± 12.9	3.9 ± 0.6	231 ± 71.8 ***	68.3 ± 14.0 ***

^1^ Number of analyzed dairy farms or sites (units of observation). ^2^ Average number of houseflies or fungal isolates (means ± SEM) per unit of observation, each consisting of the average of two fly captures as well as replicates of two fungal isolation methods. ^a–c^ Different letters indicate significant differences between means located in the same columns (dairy farms or sites) with Tukey’s honestly significant difference test (*p* < 0.05). *^,^ **^,^ *** Statistically significant difference between dry season and wet season with a *p*-value of <0.05, <0.01, <0.001, respectively. ^ns^ Not significant.

**Table 2 toxins-16-00302-t002:** Aflatoxins (AFs) and zearalenone (ZEA) produced by *Aspergillus* spp. and *Fusarium* spp. obtained from flies collected at different dairy farm sites.

Location	Number ^1^	Dry Season ^2^		Wet Season ^2^	
		*Aspergillus*	*Fusarium*	*Aspergillus*	*Fusarium*
Dairy Farm					
I	7	0.6 ^b^ ± 0.1	0.6 ^b^ ± 0.2	0.0 ^b^ ± 0.0 *	1.3 ^b^ ± 0.2 ^ns^
II	7	0.0 ^b^ ± 0.0	0.0 ^b^ ± 0.0	0.7 ^b^ ± 0.2 ^ns^	0.1 ^b^ ± 0.1 ^ns^
III	7	0.7 ^b^ ± 0.2	2.0 ^ab^ ± 0.3	0.4 ^b^ ± 0.2 ^ns^	0.7 ^b^ ± 0.2 ^ns^
IV	7	1.1 ^b^ ± 0.3	0.3 ^b^ ± 0.1	0.1 ^b^ ± 0.1 ^ns^	1.3 ^b^ ± 0.3 ^ns^
V	7	1.7 ^b^ ± 0.5	1.9 ^ab^ ± 0.4	0.0 ^b^ ± 0.0 ^ns^	3.6 ^b^ ± 0.7 ^ns^
VI	7	0.3 ^b^ ± 0.1	0.3 ^b^ ± 0.1	0.0 ^b^ ± 0.0 ^ns^	7.3 ^b^ ± 1.1 **
VII	7	0.6 ^b ±^ 0.2	1.1 ^ab^ ± 0.3	6.1 ^b^ ± 1.0 *	4.9 ^b^ ± 1.2 ^ns^
VIII	7	0.0 ^b^ ± 0.0	4.4 ^a^ ± 1.0	10.3 ^b^ ± 1.9 *	7.4 ^b^ ± 1.2 ^ns^
IX	7	0.6 ^b^ ± 0.2	1.4 ^ab^ ± 0.3	5.3 ^b^ ± 1.8 ^ns^	53.7 ^a^ ± 6.4 **
X	7	6.1 ^a^ ± 0.8	0.3 ^b^ ± 0.1	63.9 ^a^ ± 8.4 **	89.9 ^a^ ± 13.8 **
Site					
Feeders (Low)	10	0.9 ^a^ ± 0.2	0.2 ^a^ ± 0.1	12.1 ^a^ ± 3.5 ^ns^	0.4 ^a^ ± 0.2 ^ns^
Feeders (Middle)	10	0.2 ^a^ ± 0.1	0.6 ^a^ ± 0.2	1.5 ^a^ ± 0.6 *	7.9 ^a^ ± 2.4 *
Feeders (High)	10	0.2 ^a^ ± 0.1	2.0 ^b^ ± 0.5	11.4 ^a^ ± 3.0 ^ns^	20.5 ^a^ ± 3.7 *
Rearing room	10	0.8 ^a^ ± 0.2	0.1 ^a^ ± 0.1	6.8 ^a^ ± 2.0 ^ns^	8.8 ^a^ ± 1.8 *
Milking parlor	10	1.9 ^a^ ± 0.6	1.1 ^a^ ± 0.3	12.5 ^a^ ± 5.3 ^ns^	29.1 ^a^ ± 7.2 ^ns^
Silo-cutting surface	10	3.3 ^a^ ± 0.5	2.8 ^a^ ± 0.5	9.9 ^a^ ± 3.0 ^ns^	24.6 ^a^ ± 5.3 *
Feed store	10	0.9 ^a^ ± 0.2	1.8 ^a^ ± 0.4	6.6 ^a^ ± 2.4 ^ns^	27.8 ^a^ ± 8.9 ^ns^
All	70	1.2 ^b^ ± 0.3	1.2 ^b^ ± 0.3	8.7 ^a^ ± 2.8 ***	17.0 ^a^ ± 4.2 ***
AFs or ZEA-producing (No.)	70	0.3 ± 0.1	1.9 ± 0.6	5.5 ± 0.7 ***	7.2 ± 0.8 ***
AFs or ZEA-producing (%)	70	32.9	22.6	79.3	42.5
Mycotoxins (AFs or ZEA, µg/kg)	70	327 ± 145	3132 ± 1772	655 ± 98.8 ^ns^	1909 ± 817 ^ns^

^1^ Number of analyzed dairy farms and sites within dairy farms (units of observation). ^2^ Average number of houseflies or fungal isolates (means ± SEM) per unit of observation, each consisting of the average of two fly captures as well as replicates of two fungal isolation methods. ^a–b^ Different letters indicate significant differences between means located in the same columns (dairy farms or sites) with Tukey’s honestly significant difference test (*p* < 0.05). *^,^ **^,^ *** Statistically significant difference between dry season and wet season with a *p*-value of <0.05, <0.01, <0.001, respectively. ^ns^ Not significant.

**Table 3 toxins-16-00302-t003:** Significance analysis (*p*-value) of the influence of the predictor factors on the number of flies and different types of *Aspergillus* spp. and *Fusarium* spp. on dairy farms (DFs) using the generalized linear model method.

Factor	House Flies	Fungi Isolations	*Aspergillus*	*Fusarium*	AFs-Producing *Aspergillus* spp.	ZEA-Producing *Fusarium* spp.
Model	0.0000	0.0000	0.0000	0.0000	0.0000	0.0000
Station	0.0003	0.0000	0.0099	0.0095	0.0154	0.0060
Dairy farm	0.0000	0.0000	0.0000	0.0000	0.0000	0.0000
Site (DF)	0.9249	0.9816	0.9539	0.5305	0.9996	0.0025
Pluvial precipitation	0.0000	0.0000	0.0002	0.0001	0.0001	0.0000
Relative humidity	0.0010	0.0004	0.0397	0.0251	0.0553	0.0156
Coefficient of determination R^2^ (%)	74.4	67.0	46.4	46.6	42.5	53.7
Correlation coefficient with trapped houseflies (*** *p*-value < 0.001)	--	0.887 ***	0.617 ***	0.691 ***	0.630 ***	0.641 ***

**Table 4 toxins-16-00302-t004:** Meteorological conditions existing in the dairy farms during housefly capture *.

Meteorological Variables	A. Dry Season		B. Wet Season		Mean Increase
	Mean	STD	Mean	STD	(B − A)/A%
Pluvial precipitation (mm/mo)	0.24 ^b^	±0.37	48.3 ^b^	±31.6	20,025
Relative humidity (%)	43.0 ^b^	±5.6	67.7 ^b^	±1.44	57.4
Middle temperature (°C)	18.4 ^b^	±0.64	19.5 ^b^	±0.06	6.0
Maximum temperature (°C)	28.3 ^b^	±0.69	26.1 ^b^	±0.14	−7.8
Minimum temperature (°C)	8.5 ^b^	±1.41	13.0 ^b^	±0.21	52.9

* Monthly average of the daily record in the National Network of Automated Agrometeorological Stations (https://clima.inifap.gob.mx/lnmysr/Estaciones/Mapa (accessed on 28 June 2024)).

## Data Availability

The data from the study are included in this paper.

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
