# Peer review of "Increased Dissemination of Aflatoxin- and Zearalenone-Producing Aspergillus spp. and Fusarium spp. during Wet Season via Houseflies on Dairy Farms in Aguascalientes, Mexico"

_toxins, 2024, doi:10.3390/toxins16070302_

Round 1

Reviewer 1 Report

Comments and Suggestions for Authors

Manuscript ID: Toxins-3060299

Title: Increased dissemination of aflatoxins- and zearalenone- producing Aspergillus spp. and Fusarium spp. during wet season via houseflies on dairy farms in Aguascalientes, Mexico.

General Observations:

The current manuscript reports an observational study from dairy farms in a limited area of the Mexican Plateau.  The study evaluated the effects of the rainy season weather conditions on the prevalence of Aspergillus and Fusarium fungi and their associated toxins (aflatoxin and zearalenone) and the increase in the fly population compared to the dry season on ten dairy farms. The study observed a large difference in the fly population between the two seasons, which correlated with increase of the two main fungi of interest and their mycotoxins.  The authors hypothesize that the flies act as a vector for spore dispersal of both toxigenic fungi which contaminate the feed and subsequently the milk production, thereby threatening the health of the cows and of human consumers.  Therefore, it is important to control the flies in order to decrease the spread of mycotoxins especially during the rainy season. Although there has always been a connection between insects and mycotoxin infection, flies were seldom considered relevant and appear to be novel in this context.

Although the locations of the farms were not randomly chosen, they appear to represent the diversity of dairy farm practices in the target area. 

The manuscript is well written, requiring few edits and minor changes in phrasing for clarity.  The introduction is well organized and contains relevant references.  Methods are well described, and analyses appear appropriate. The general observations and statistical analyses of the data appear to be in agreement. It would be useful to have graphs, in addition to the tables, for a visual presentation of the data.

Specific Comments:

Line 6-7: should read… distribution; their mycotoxins, aflatoxins (AF) and zearalenone (ZEA)…

Line 10: should read…in the semi-arid…

Line 11: should read…points on each farm…

Line 16: should read… compared to the dry season there was a higher abundance of flies as well as AFs- and ZEA-producing

Line 17: should read…in the wet season

Line 18-19: should read… spread of AF- and ZEA-producing Aspergillus spp. and Fusarium spp. through houseflies and incorporate their mycotoxins into the food chain.

Line 22: should read… were associated with a significant increase in both houseflies and…

Line 23: should read…fungal populations.

Line 24: should read…on dairy farms…

Line 25: should read…health. Moreover,…

Line 28: should read…with mycotoxins…

Line 119: The Result section would benefit from graphical presentation of the data

Line 163: should read…a highly significant…

Line 198: should read…fungal populations.

Line 199: should read…on dairy farms…

Line 207: should read…on dairy farms…

Line 213: should read…a great impact…

Line 215-217: please rephrase sentence for clarity

Line 258-259: should read…was also increased.

Line 272: should read…on dairy farms.

Line 274: Conclusion section consists of two very long sentences. Please break them up.

Line 316: should read…during the two seasons…

Line 327-328: Please rephrase sentence for clarity

Comments on the Quality of English Language

The manuscript is well written, requiring few edits and minor changes in phrasing for clarity.  

Author Response

Reviewer: The current manuscript reports an observational study from dairy farms in a limited area of the Mexican Plateau.  The study evaluated the effects of the rainy season weather conditions on the prevalence of Aspergillus and Fusarium fungi and their associated toxins (aflatoxin and zearalenone) and the increase in the fly population compared to the dry season on ten dairy farms. The study observed a large difference in the fly population between the two seasons, which correlated with increase of the two main fungi of interest and their mycotoxins.  The authors hypothesize that the flies act as a vector for spore dispersal of both toxigenic fungi which contaminate the feed and subsequently the milk production, thereby threatening the health of the cows and of human consumers.  Therefore, it is important to control the flies in order to decrease the spread of mycotoxins especially during the rainy season. Although there has always been a connection between insects and mycotoxin infection, flies were seldom considered relevant and appear to be novel in this context. Although the locations of the farms were not randomly chosen, they appear to represent the diversity of dairy farm practices in the target area. The manuscript is well written, requiring few edits and minor changes in phrasing for clarity.  The introduction is well organized and contains relevant references.  Methods are well described, and analyses appear appropriate. The general observations and statistical analyses of the data appear to be in agreement.              

Reviewer: Specific Comments: It would be useful to have graphs, in addition to the tables, for a visual presentation of the data.

Authors: A graph was added that compiles the main findings of the study and compares the evolution of fungal and fly populations in the dry and wet seasons (Lines: 200-204).

Reviewer: Line 6-7: should read… distribution; their mycotoxins, aflatoxins (AF) and zearalenone (ZEA)…

Authors: The paragraph was rewritten to improve their comprehension (Lines: 6-7)

Reviewer: Line 10: should read…in the semi-arid…

Authors: The phrase arrangement was performed (Line: 10).

Reviewer: Line 11: should read…points on each farm…

Authors: The phrase arrangement was performed (Line: 11).

Reviewer: Line 16: should read… compared to the dry season there was a higher abundance of flies as well as AFs- and ZEA-producing

Authors: The phrase arrangement was performed (Line: 16).

Reviewer: Line 17: should read…in the wet season

Authors: The phrase arrangement was performed (Line: 17).

Reviewer: Line 18-19: should read… spread of AF- and ZEA-producing Aspergillus spp. and Fusarium spp. through houseflies and incorporate their mycotoxins into the food chain.              

Authors: The phrase arrangement was performed (Lines: 19-20).

Reviewer: Line 22: should read… were associated with a significant increase in both houseflies and…

Authors: The phrase arrangement was performed (Line: 23).

Reviewer: Line 23: should read…fungal populations.

Authors: The phrase arrangement was performed (Line: 24).

Reviewer: Line 24: should read…on dairy farms…

Authors: The phrase arrangement was performed (Line: 25).

Reviewer: Line 25: should read…health. Moreover,…

Authors: The phrase arrangement was performed (Line: 26).

Reviewer: Line 28: should read…with mycotoxins…

Authors: The phrase arrangement was performed (Line: 29).

Reviewer: Line 119: The Result section would benefit from graphical presentation of the data

Authors: A graph was added that compiles the main findings of the study and compares the evolution of fungal and fly populations in the dry and wet seasons (Lines: 200-204).

Reviewer: Line 163: should read…a highly significant…

Authors: The phrase arrangement was performed (Line: 185).

Reviewer: Line 198: should read…fungal populations.

Authors: The phrase arrangement was performed (Line: 239).

Reviewer: Line 199: should read…on dairy farms…

Authors: The phrase arrangement was performed (Line: 240).

Reviewer: Line 207: should read…on dairy farms…

Authors: The phrase arrangement was performed (Line: 207).

Reviewer: Line 213: should read…a great impact…

Authors: The phrase arrangement was performed (Line: 248).

Reviewer: Line 215-217: please rephrase sentence for clarity

Authors: The sentence was rewritten to improve their comprehension (Lines: 256-258).

Reviewer: Line 258-259: should read…was also increased.

Authors: The phrase arrangement was performed (Lines: 299-300).

Reviewer: Line 272: should read…on dairy farms.

Authors: The phrase arrangement was performed (Line: 314).

Reviewer: Line 274: Conclusion section consists of two very long sentences. Please break them up.

Authors: The conclusion section was break up to improve their comprehension (Lines: 323-324).

Reviewer: Line 316: should read…during the two seasons…

Authors: The phrase arrangement was performed (Line: 358).

Reviewer: Line 327-328: Please rephrase sentence for clarity

Authors: The phrase arrangement was performed (Lines: 369-371).

Reviewer 2 Report

Comments and Suggestions for Authors

The topic of the work is interesting, but the work needs to be thoroughly revised before publication. The statements are often too general and could lead to misinterpretation.

Here are my detailed comments:

In the abstract and in the introduction, the authors state that “Aspergillus spp. and Fusarium spp. are toxigenic fungi...”, which is not entirely correct. There are species of the genus Aspergillus that are not mycotoxigenic. It should be worded more clearly, such as “some (or many or various) species of the genera Aspergillus and Fusarium are mycotoxicgenic”. AFs and ZEA are classified as mycotoxins and this term should be used.

Also, the authors claim at the beginning of the introduction that “AFs and ZEA are toxic compounds ....”, it would be better to say that they are mycotoxins and then give a definition of mycotoxins. The same paper can be cited, which also contains the definition of mycotoxins.

The toxicity of AFs and ZEA should be adequately discussed in the introduction, now the Afs toxicity part of the discussion. AFs are considered among most toxic mycotoxins, which should be clear to the reader.

The authors discuss the house fly as a possible vector for the spread of conidia of mycotoxin-producing fungi. To better understand the risk, it would be good to explain on which raw materials AFs and ZEA-producing fungi grow.

It would be highly desirable to explain/discuss in the introduction why and how the housefly could increase the risk of mycotoxin occurrence by transporting conidia.

Tables 1 and 2 should be better explained, what is No? is it the number of flies examined or what?

The same applies to Figure 3.

In the introduction, the authors correlate the occurrence of mycotoxigenic fungi and flies with weather conditions, and the data confirm this. What is new about this data? There should be an analysis of the species that occur in the dry and wet seasons. Fungi should be identified to species.

In the introduction it is said that the fly can transport conidia over 100 km. It would be interesting to analyse the fungal population in situ and that isolated from flies.

Please pay attention when formulating sentences and avoid structures that allow the conclusion conidia = mycotoxin.

Comments on the Quality of English Language

Moderate editing of English language required

Author Response

Reviewer: The topic of the work is interesting, but the work needs to be thoroughly revised before publication. The statements are often too general and could lead to misinterpretation. Here are my detailed comments:

Reviewer: In the abstract and in the introduction, the authors state that “Aspergillus spp. and Fusarium spp. are toxigenic fungi...”, which is not entirely correct. There are species of the genus Aspergillus that are not mycotoxigenic. It should be worded more clearly, such as “some (or many or various) species of the genera Aspergillus and Fusarium are mycotoxicgenic”. AFs and ZEA are classified as mycotoxins and this term should be used.      

Authors: The abstract and introduction were rewritten to improve their comprehension (Lines: 6-7 and 35-36).

Reviewer: Also, the authors claim at the beginning of the introduction that “AFs and ZEA are toxic compounds ....”, it would be better to say that they are mycotoxins and then give a definition of mycotoxins. The same paper can be cited, which also contains the definition of mycotoxins.

Authors: A definition of mycotoxins was elaborated based on the reviewer and the suggested bibliographic reference (Lines: 32-37).

Reviewer: The toxicity of AFs and ZEA should be adequately discussed in the introduction, now the Afs toxicity part of the discussion. AFs are considered among most toxic mycotoxins, which should be clear to the reader.  

Authors: The toxicity of AFs and ZEA was paraphrased, and information was added (highlighted text) on their toxic and carcinogenic capacity (Lines: 48-62).

Reviewer: The authors discuss the house fly as a possible vector for the spread of conidia of mycotoxin-producing fungi. To better understand the risk, it would be good to explain on which raw materials AFs and ZEA-producing fungi grow.

Authors: It was pointed out in which crops the FAs and ZEA-producing fungi grow, and it was also specified in which feeds destined to dairy cows these fungi usually develop (Lines: 39-40 and 84-87).

Reviewer: It would be highly desirable to explain/discuss in the introduction why and how the housefly could increase the risk of mycotoxin occurrence by transporting conidia.

Authors: The risk that the increased dissemination of conidia by the house fly represents for the contamination of the food chain was discussed (Lines: 121-129).

Reviewer: Tables 1 and 2 should be better explained, what is No? is it the number of flies examined or what?

Authors: Tables 1 and 2 were better explained; headings were clarified and notes were specified (Lines: 140-143 and 173-175).

Reviewer: The same applies to Figure 3.

Authors: Table 3 was also explained (Lines: 206-209).

Reviewer: In the introduction, the authors correlate the occurrence of mycotoxigenic fungi and flies with weather conditions, and the data confirm this. What is new about this data?       

Authors: The research problem statement was rewritten to highlight that, although the effect of climate on fungal and fly populations is known, the combined effect of the concurrent growth of both populations and the risk caused to dairy and food chain contamination by MTs is not adequately known. (Lines: 121-128)

Reviewer: Please pay attention when formulating sentences and avoid structures that allow the conclusion conidia = mycotoxin.        

Authors: The entire manuscript was revised to avoid directly associating the dissemination of conidia with mycotoxin contamination. The wording was adjusted where necessary (Lines: 127-129, 281-285 and 319-321).

Reviewer: In the introduction it is said that the fly can transport conidia over 100 km. It would be interesting to analyze the fungal population in situ and that isolated from flies.             

Authors: The authors agree with this observation. We propose to study the molecular mechanism of identification of some type of organism (e.g., a non-toxigenic A. flavus L-morphotype. Cfr. https://doi.org/10.3390/toxins14070437) and evaluate its field dissemination by the house fly.

Reviewer: There should be an analysis of the species that occur in the dry and wet seasons. Fungi should be identified to species.         

Authors: The authors agree with this statement. However, in the present study, a total of 5,051 fungal isolates were identified from flies collected in both the dry and wet seasons (693 Aspergillus isolates, 1,274 Fusarium and 3084 of other genera). This was a large amount of work. Therefore, it was not possible to process everything simultaneously; however, the isolates have been adequately preserved to finish a little later, given that the speed of this process depends on the rate of delivery of public and institutional resources. We consider that processing all the isolates could take about one additional year, so we preferred not to delay the communication of our findings while we can conclude the morphologic, toxigenic and molecular characterization of the large population of fungal isolates.

Reviewer 3 Report

Comments and Suggestions for Authors

It is a nice descriptive study regarding the correlation between houseflies and Aspergillus and Fusarium spp. on dairy farms under different environmental conditions. The authors started with a study design, collection of houseflies, isolation of fungal species, and mycotoxin quantitation, followed by data analysis to identify the correlation between houseflies and identified fungal species under dry/wet meteorological patterns. In general the data are comprehensive and of acceptable quality. The study itself would lead to useful information for mycotoxin management so I do not have any major objections to the publication of the study. In the future studies it would be worth measuring the mycotoxin occurrence and concentrations of AFs, ZEN and other mycotoxins in the animal feed. This way the authors could further demonstrate the impact of houseflies and weather patterns on the diary farms. It would also be interesting to see if the level of AFM1 in milk would be affected during dry/wet seasons. If the data are available, the analysis of mycotoxins should include QC data (lines 362-376) to demonstrate method performance. Table 3: not sure how the authors minimized the impact of the sampling of houseflies.       

Author Response

Reviewer: It is a nice descriptive study regarding the correlation between houseflies and Aspergillus and Fusarium spp. on dairy farms under different environmental conditions. The authors started with a study design, collection of houseflies, isolation of fungal species, and mycotoxin quantitation, followed by data analysis to identify the correlation between houseflies and identified fungal species under dry/wet meteorological patterns. In general the data are comprehensive and of acceptable quality. The study itself would lead to useful information for mycotoxin management so I do not have any major objections to the publication of the study.    

Reviewer: In the future studies it would be worth measuring the mycotoxin occurrence and concentrations of AFs, ZEN and other mycotoxins in the animal feed. This way the authors could further demonstrate the impact of houseflies and weather patterns on the diary farms.

Authors: Quantification results of AFs and ZEA in the total mixed ration samples collected during the same time as the fly capture were added. (Lines: 226-235, 414-418).

Reviewer: It would also be interesting to see if the level of AFM1 in milk would be affected during dry/wet seasons.

Authors: AFM1 quantification results in raw milk samples collected during the same time as fly trapping were added. (Lines: 226-235, 414-418). The other MTs data agreed with the increase reported for flies and fungi

Reviewer: If the data are available, the analysis of mycotoxins should include QC data (lines 362-376) to demonstrate method performance.

Authors: Quality control data were added for the quantification of MTs (Lines: 422-428).

Reviewer: Table 3: not sure how the authors minimized the impact of the sampling of houseflies. Authors: Table 3 was better explained (Lines: 206-209).

Reviewer 4 Report

Comments and Suggestions for Authors

This is a very interesting study, for the first time revealing that rainy weather conditions on dairy farms increase spread through houseflies of AF- and ZEN-producing Aspergillus spp. and Fusarium spp. and incorporation of their mycotoxins into food chain. The study will benefit readers focusing on mycotoxins control. The current manuscript was well-written and could be considered for publication in Toxins after only minor revision.

1、 As for quantification of mycotoxins, the detection limit and linear range should be given.

2、The isolated fungi were identied by morpholoigcal characteristics in this study. Apart from Aspergillus and Fusarium presnted in Fig 1, ohter fungi including Mucor, Rizhapus, Penicillium, Cladosporium, Absidia, Cunninghamella, Eurotium, Geotrichu, and Alternaria should also be prestented in supplementary files.

3、Fusarium spp. could produce ZEN and also Deoxynivalenol(DON) one of the most notorious mycotoxins causing worldwide attention, why DON was not detected in this study.

4、The unit for table 1 and 2 was not clear. Morever, P should be italic throughout the manuscript.

Author Response

Reviewer: This is a very interesting study, for the first time revealing that rainy weather conditions on dairy farms increase spread through houseflies of AF- and ZEN-producing Aspergillus spp. and Fusarium spp. and incorporation of their mycotoxins into food chain. The study will benefit readers focusing on mycotoxins control. The current manuscript was well-written and could be considered for publication in Toxins after only minor revision.              

Reviewer: 1. As for quantification of mycotoxins, the detection limit and linear range should be given.

Authors: The detection limit and linear range for quantification of mycotoxins and quality control data were added (Lines: 421-428).

Reviewer: 2. The isolated fungi were identied by morpholoigcal characteristics in this study. Apart from Aspergillus and Fusarium presnted in Fig 1, ohter fungi including Mucor, Rizhapus, Penicillium, Cladosporium, Absidia, Cunninghamella, Eurotium, Geotrichu, and Alternaria should also be prestented in supplementary files.

Authors: The authors agree with this statement. In the present study, a total of 5,051 fungal isolates were identified; but the images currently available are not of the quality that readers of the journal Toxins deserve; so, we extend a sincere apology for not addressing this appropriate and pertinent observation.

We consider that morphologic, toxigenic and molecular characterization process could take about one additional year, so we preferred not to delay the communication of our findings while we can conclude the whole characterization of the large population of fungal isolates.

Reviewer: 3. Fusarium spp. could produce ZEN and also Deoxynivalenol (DON) one of the most notorious mycotoxins causing worldwide attention, why DON was not detected in this study. Authors: We collected 1,274 Fusarium isolates. This was a large amount of work; therefore, it was not possible to process all non-regulated mycotoxins simultaneously. However, the isolates have been adequately preserved to finish a little later, given that the speed of this process depends on the rate of delivery of public and institutional resources.

Reviewer: 4. The unit for table 1 and 2 was not clear.

Authors: Tables 1 and 2 were better explained; headings were clarified, and notes were specified (Lines: 140-143 and 173-175).

Reviewer: Morever, P should be italic throughout the manuscript.

Authors: The complete manuscript was revised and the style of the P-value was changed to italics.

Round 2

Reviewer 2 Report

Comments and Suggestions for Authors

There are some improvements in the paper after the revision but I still think it is not ready for publication. The topic and results are interesting but some improvements in the structure and content should be done.

The paper deals with the housefly as a possible vector for dissemination of conidia of mycotoxigenic fungi. The introduction chapter should present well the topic, but there is still space for improvement in the introduction chapter of this paper. The authors choose to analyse the presence of the conidia of two different mycotoxigenic genera on houseflies. The choice is well explained, being the mycotoxins of these two genera most common in Mexican food. The authors affirm that the presence of the conidia of these genera presents a risk, but this risk is described very generically. In some points it seams that already the presence of conidia equals the presence of a mycotoxin. The two mycotoxins have very different toxicity, AFB1 is classified as a cancerogenic while ZEA is not considered as cancerogenic. Nevertheless both can present a risk for animal and human health and impair the animal production (partially explained in the introduction) and therefor the risk of their major dissemination by horsefly, as an important part for this research, shod be better explained. The two chosen genera have different growing conditions and behave differently. While Fusarium spp mostly contaminate crops in the field, Aspergillus spp, due to the possibility to grow on lower aw, can contaminate also during the storage. AFs can be produced even in mangers if not consumed feed has been left, and enter the food chain with the next feedings. Generally the major intake of the AF and ZEA in animals s due to cereal based feed (for AFs mainly corn, and peanuts) The authors should briefly address these topics. Furthermore, it would be good to explain to the readers in which growing phase are the cereals in the wet season (I don’t know it, for example). This could be an additional risk factor. It would be good to address the risk for pre- and postharvest period.

Please rearrange the introduction and discussion to make them more representative of the importance of this research.

Comments on the Quality of English Language

The English language is fine, some minor changes might be necessary

Author Response

Reviewer: There are some improvements in the paper after the revision but I still think it is not ready for publication. The topic and results are interesting but some improvements in the structure and content should be done. The paper deals with the housefly as a possible vector for dissemination of conidia of mycotoxigenic fungi. The introduction chapter should present well the topic, but there is still space for improvement in the introduction chapter of this paper. The authors choose to analyse the presence of the conidia of two different mycotoxigenic genera on houseflies. The choice is well explained, being the mycotoxins of these two genera most common in Mexican food.

Reviewer: The authors affirm that the presence of the conidia of these genera presents a risk, but this risk is described very generically.

Authors: The risk that the increased dissemination of conidia by the house fly represents for the contamination of the food chain was discussed (Lines: 121-129). In addition, the research problem statement was rewritten to highlight that, although the effect of climate on fungal and fly populations is known, the combined effect of the concurrent growth of both populations and the risk caused to dairy and food chain contamination by MTs is not adequately known. (Lines: 121-128)

Reviewer: In some points it seams that already the presence of conidia equals the presence of a mycotoxin.

Authors: The entire manuscript was revised to avoid directly associating the dissemination of conidia with mycotoxin contamination. The wording was adjusted where necessary (Lines: 127-129, 281-285 and 319-321).

Reviewer: The two mycotoxins have very different toxicity, AFB1 is classified as a cancerogenic while ZEA is not considered as cancerogenic. Nevertheless both can present a risk for animal and human health and impair the animal production (partially explained in the introduction) and therefor the risk of their major dissemination by horsefly, as an important part for this research, shod be better explained.

Authors: The toxicity of AFs and ZEA was paraphrased, and information was added (highlighted text) on their toxic and carcinogenic capacity (Lines: 48-62).

Reviewer: The two chosen genera have different growing conditions and behave differently. While Fusarium spp mostly contaminate crops in the field, Aspergillus spp, due to the possibility to grow on lower aw, can contaminate also during the storage. AFs can be produced even in mangers if not consumed feed has been left, and enter the food chain with the next feedings. Generally the major intake of the AF and ZEA in animals s due to cereal based feed (for AFs mainly corn, and peanuts) The authors should briefly address these topics.

Authors: It was pointed out in which crops the FAs and ZEA-producing fungi grow, and it was also specified in which feeds destined to dairy cows these fungi usually develop (Lines: 39-40, 84-87 and 261-264).

Reviewer: Furthermore, it would be good to explain to the readers in which growing phase are the cereals in the wet season (I don’t know it, for example). This could be an additional risk factor. It would be good to address the risk for pre- and postharvest period.

Authors: A precision on the time of cultivation and harvesting of forages produced on the farms, as well as other feed ingredients, was included (Lines 84-87 and 347-351).

Reviewer: Please rearrange the introduction and discussion to make them more representative of the importance of this research.

Authors: As mentioned in the previous sections, the introduction and discussion sections of the manuscript were rearranged.

Round 3

Reviewer 2 Report

Comments and Suggestions for Authors

The paper is of interest to readers, and my comments aim to improve its quality and impact. I think that the introduction needs to be restructured and not just a few sentences added. The problem is that a housefly is not a plant pest and therefore is not a direct vector for mycotoxigenic fungi and will not contribute directly to fungal colonization in the field. However, there could be an indirect contribution to contamination in the field and a direct contribution to colonization in the post-harvest period. The introduction should explain HOW and WHY the presence of conidia of mycotoxigenic fungi on flies could increase the risk of mycotoxin occurrence in feed and food. The presence of conidia does not directly imply the presence of mycotoxin (nor does fungal growth).Therefore, I proposed to explain where the occurrence of these two mycotoxins was reported. The authors cite oilseeds by-products as a component of the feed. If one explains beforehand that the occurrence of AF has been reported in oilseeds (and that the concentration in oilseeds is generally much higher than in starch seeds), the risk becomes clearer. Houseflies could also directly promote the occurrence of AF on human foods (e.g. dried fruits). For this reason, I have proposed to summarize a brief overview of both mycotoxins (toxicity, occurrence, producers, environmental conditions for their growth) in a few sentences. This information should mainly be included in the introduction, something can develop in the discussion, but it certainly should not be included in Material and Methods. The information I have now mentioned is (partially) reported in part of the text, but should be organized in a clear structure that allows a reader with less experience with mycotoxins to follow it and understand the aim and hypothesis of the paper.

Comments on the Quality of English Language

English is fine

Author Response

Reviewer: The paper is of interest to readers, and my comments aim to improve its quality and impact. I think that the introduction needs to be restructured and not just a few sentences added. 

Authors: In accordance with the observations of reviewer, the organization of the introduction section of the manuscript was completely restructured.

Reviewer: For this reason, I have proposed to summarize a brief overview of both mycotoxins (toxicity, occurrence, producers, environmental conditions for their growth) in a few sentences.

Authors: In accordance with the observations of reviewer, in this new version we explain the general characteristics of the AFs and ZEA producing fungi, their occurrence, the environmental conditions for their growth and the toxicity of these two mycotoxins reported. Furthermore, the association between the dissemination of mycotoxigenic fungal conidia through flies and the increased risk of mycotoxin appearance in feed and food was highlighted.